



# The characterization of long-range transported North American biomass burning plumes: what can a multi-wavelength Mie-Raman-polarization-fluorescence lidar provide?

Qiaoyun Hu[1], Philippe Goloub[1], Igor Veselovskii[2], and Thierry Podvin[1]

[1]Univ. Lille, CNRS, UMR 8518 - LOA - Laboratoire d'Optique Atmosphérique, F-59650 Lille, France
[2]Prokhorov General Physics Institute of the Russian Academy of Sciences, Moscow, Russia

**Correspondence:** Qiaoyun Hu (qiaoyun.hu@univ-lille.fr)

**Abstract.** This article presents a study of long-range transported biomass burning aerosols (BBA) originated from the North American wildfires in September 2020. The BBA plumes presented in this study were in the troposphere and underwent 1-2 weeks aging before arriving at the observations site. A novel dataset $2\alpha+3\beta+3\delta+\phi$ ($\alpha$: extinction coefficient, $\beta$: backscatter coefficient, $\delta$: particle linear depolarization ratio, PLDR, $\phi$: fluorescence capacity) derived from lidar observations is provided

for the characterization of long-range transported BBA. The fluorescence capacity describes the ability of aerosols in producing fluorescence when exposed to UV excitation. In the observations of BBA episode, plumes from different wildfire activities have been characterized. In the BBA plumes, we detected low PLDRs, i.e. lower than 0.03 at all wavelengths, as well as enhanced PLDRs ($\text{PLDR}_{355,532,1064} \approx 0.15 - 0.18, 0.12 - 0.14, 0.01 - 0.02$) with a similar spectral dependence that had been observed in the aged BBA plumes in the upper troposphere and lower stratosphere (Canadian smoke in 2017 and Australian smoke in

2019-2020). Obvious variations in Angström exponent (-0.3 – 1.5), lidar ratios (20–50 sr at 355 nm, 42–90 sr at 532 nm) and fluorescence capacity ($1.0\times10^{-4}$ – $4.0\times10^{-4}$ ) are also observed during the BBA episode. These variations are coupled with the variation of altitudes, water vapor content and wildfire events. It reflects that the properties of aged BBA particles are highly varied and depend on complex mechanisms, such as burning process and the aging process. The results also pointed out the inhomogeneity of the aging process in the BBA plumes, which means that particles in the core of the plume aged differently

with those at the plume edge due to the impact of water vapor, temperatures, particle concentration and so on. These chemical and physical processes involved in BBA aging and how they could impact the particle properties are not yet well understood. In addition, our observations identified the ice crystals mixing with BBA particles, which indicates that BBA could act as ice nucleating particles (INP) at tropospheric conditions. The lidar fluorescence proves to be an efficient tool in studying the interaction of clouds and BBAs due to its high sensitivity. Recent studies claimed that aged BBA particles are more effective

INPs than they were thought. As BBAs are becoming an important atmospheric aerosols with growing global wildfires, our observations could improve our characterization about aged BBA particles and the understanding of their importance in ice cloud formation.



# 1 Introduction

Severe wildfires caught the attention of the public and the scientific communities in resent years. Wildfires in North America
increase in both frequency and intensity, due to the warming-up of the climate. Regarding to the global scale, the duration of
fire seasons increased by about 19% from 1979 to 2013 (Jolly et al., 2015; Schoennagel et al., 2017). Wildfires can directly
impact the vegetation of earth's surface, the fertility of soils, water cycle and human society (Santín and Doerr, 2016; Ditas
et al., 2018; May et al., 2014; O'Dell et al., 2020). Copious amount of fire emissions, including gases and particles, were
injected into the troposphere and occassionally into the upper troposphere and lower stratosphere (UTLS). Those emissions
pose threats to the air quality and human health, because they contain mainly fine particles and some chemical compounds
in wildfire emissions are toxic. They could alternate the planetary radiation budget of the planet by scattering and absorbing
the incoming solar radiation and influence the cloud process by modifying cloud properties and acting as INP (ice nucleation
particle) and CCN (cloud condensation nuclei).

Wildfire emissions are composed of a mixture of gases, e.g. carbon monoxide, polycyclic aromatic hydrocarbons (PAHs) con-
taining gas-phase hazardous air pollutants, water vapor and aerosol particles, such as black carbon (BC), organic carbon (OC)
etc. In addition, water soluble ions (potassium, halides, sulfates) can also be produced by the decomposition of biomass and
the subsequent condensation in the smoke plume. The OC components account for a substantial fraction, about 20% to 90%
in wildfire emissions, much higher than BC. Humic-like substances, which are a complex group of high molecular weight
organic compounds, contribute considerably to the mass of organic compounds and influence the light absorbing properties of
aerosols (Urbanski et al., 2008; Santín and Doerr, 2016; Wu et al., 2018). For convenience, we use hereafter BBA (biomass
burning aerosols) to refer to the aerosols resulting from wildfire emissions. Due to the complexity in the burning and the aging
processes, the BBA properties are highly variable. The initial composition and size distribution of BBA depends on the burning
materials (i.e vegetation type, such as grassland, boreal forest, etc) and environmental conditions (temperature, moisture, soil
properties, etc), which impact the combustion efficiency and the burning processes (i.e. flaming and smoldering). The smol-
dering phase, which is with lower temperature, could lead to a substantially higher conversion of burning materials to toxic
compounds than the flaming phase, and produce more weakly absorbing OC aerosols. Fire emissions from high temperatures
and the flaming phase tend to contain increased proportion of BC to OC and have smaller smoke particles (Reid et al., 2005;
Garofalo et al., 2019).

Fresh BBAs are fractal like aggregates of BC cores and OC coatings (China et al., 2013, 2015). During long-range trans-
port, BBAs undergo complex aging processes, including condensation, oxidation, evaporation, coagulation, compaction and
humidification. These processes could modify the particles' morphology, composition, optical and chemical properties and hy-
groscopic properties. The aging process of wildfire emissions is not yet adequately characterized. It includes a complex group
of chemical reactions and physical changes, among which the gas-to-particle condensation driven by oxidation and evaporation
driven by aerosol dilution, are two important and competing processes. The oxidation process can produce more organic matter
with lower volatility, while it could also lead to increased volatile SOA (secondary organic aerosol) due to fragmentation of
some macromolecules. The evaporation tends to decrease the partitioning of OC aerosols as dilution shifts the gas-particle





equilibrium more to the gas end. Measurements in laboratories and field campaigns do not reach a consensus on the relative balance of the two competing processes as they are controlled by many factors, such as the dilution rate, exposure to oxidants (O3, OH radical), travel time and so on. The growth of particle size is found in both near-field and longtime aged BBAs, while the causes might not be the same. Condensational growth is the dominating factor in fresh and near-field BBA, while coagulational growth is dominating in aged BBA. Abel et al. (2003) reported a increase of SSA (single scattering albedo) from about 0.84 to 0.90 in 5 hours after the emission, which is likely due to the condensation of scattering organic components on absorbing soot particles during early stage aging. Similar changes in SSA are derived by Kleinman et al. (2020) when investigating near-field BBA plumes from western US. The aging process can result in complex BBA structures and varying properties. Dahlkötter et al. (2014) sampled BBA particles in a 3-4 days old plume from western US wildfires in the UTLS. They found that a significant proportion of the particles are BC-containing particles mixed with light-scattering materials and the mixing state is highly varying. Observations of long-ranged transported Canadian smoke in 2017 and Australian smoke in 2019-2020 in UTLS showed enhanced PLDRs (particle linear depolarization ratio) that had never been detected in tropospheric smoke layers (Haarig et al., 2018; Hu et al., 2019; Ohneiser et al., 2020). Similar signatures of PLDRs have also been observed in aged tropospheric BBAs originated from wildfires (Murayama et al., 2004; Burton et al., 2015). Researchers thought such characteristics of PLDRs are indications of complicated BBA particle morphology and effects of longtime aging. Since then, many efforts have been done to simulate aged BBA particles' optical properties. The simulations indicate that aerosol properties, such as the absorption and PLDRs, are strongly dependent on the morphology, mixing state and fractions of different components.

BBAs contribute to ice nucleating (IN) mainly through immersion freezing, contact freezing , deposition freezing, as well as pore condensation and freezing(Kanji et al., 2011; Umo et al., 2015; Grawe et al., 2016; Kanji et al., 2017; Umo et al., 2019). These mechanisms are not yet well resolved and data about the onset condition of ice nucleation by BBA particles are diverse. As wildfire emissions become a more and more important atmospheric aerosol source, the role of BBAs as INP becomes increasingly important and different theories have been proposed about how the BBA impact the activation of INP. One important argument is that the amorphous state of organic matter and complicated morphology of BBAs at atmospheric conditions could influence the ability in water diffusion, thus causing strong perturbations in the prediction of cloud formation (Berkemeier et al., 2014). Recently, Jahn et al. (2020) reported a different mechanism. They claimed that the inorganic elements naturally presented in biomass could be transformed by combustion process to minerals, which are very ice active over a wide range of temperature. Studies also found that the aging process could potentially increase the capability of BBA in cloud formation (Lupi and Molinero, 2014; Jahl et al., 2021). The characterization of BBA properties in light scattering and cloud formation after long-range transport is an important direction to improve the representation of aerosol model and further, the accuracy of the climate model.

The 2020 California wildfires are the largest in the Californian modern history. They comprise about 9639 fire activities reported in the California state, located in the western coast of the United States. The fire season started from the beginning of May 2020. By the end of the year, nearly 10,000 fires had burned over 4.2 million acres, more than 4% of the state's roughly 100 million acres of land (https://www.fire.ca.gov/incidents/2020/). In mid-August, the first 'gigafire' spreading over 7 coun-





ties in northern California was recorded. 'Gigafire' is a level above 'megafire' and describes a blaze burning at least a million acres of land. Satellite observations indicated this fire activity caused huge BBA emissions over the middle and western US, while the plumes faded away after 1 week without being transported to other continents. In September 2020, the fire activities

revived after a short pause . Fire emissions were lofted into UTLS by high winds or convective pyro-cumulonimbus, and then transported to Europe by the prevailing westerlies. The most dense smoke plumes detected at the observatory ATOLL (ATmospheric Observatory of LiLle) in Lille is in the period of 10-23 September. In this study, we analyzed the observations from multiple instruments and the optical properties derived from a multi-wavelength Mie-Raman polarization lidar equipped with a fluorescence channel.

## 2   Observations

### 2.1   Satellite observations – OMPS and CALIPSO

According to satellite observations, the smoke plumes detected in Lille on 10-22 September are attributed to the fire activities on 04-11 September in California and Oregon state in the USA. On 04-06 September 2020, the Creek fire started in the southwestern California. The Creek fire is the most intense wildfire in September 2020. On 05 September, the fires generated

a pyro-cumulonimbus (PyCb) cloud reaching about 16000 m, which might be the highest PyCb cloud ever (Creek Fire, 2020). The plumes from Creek fire are obvious in the earth's true color image (Figure 1(a) ) and the maps of the UVAI from OMPS observations shown in Supplementary information (SI). The center of the lofted plumes was at (37.7N, 117.8W) on 06 September and then transported rapidly toward the northeast. On 07-10 September, the forest in the west coast of California and Oregon started to burn, which will be called Oregon fires hereafter (Oregon Fire, 2020). The burning area was in serious drought and

under the blasts of fierce winds, which lifted vast fire emissions into the atmosphere. On 09 September, the wind whipped up fires and generated a pyro-cumulus cloud. Due to cyclone activities, the plumes were trapped on the eastern Pacific ocean and west coast of the US on 07-11 September, as shown in Figure 1(b). The plumes started spreading on 12 September and arrived at Europe on 16-18 September.

Figure 3 shows the CALIPSO observations in the transport pathway. The plumes in Figure 3(a, b, c) and in Figure 3(d, e,

f) are attributed to the Creek fire and the Oregon fire, respectively. The plume origins were identified using HYSPLIT back trajectories and OMPS UVAI maps, shown in SI. The plumes emitted by the Creek fires are mostly classified as 'elevated smoke' by the CALIPSO aerosol typing algorithm (Version 4.21), while the plumes emitted from Oregon fires are mostly detected as 'polluted dust'. The reason is that the particle linear depolarization ratio at 532 nm ($PLDR_{532}$) of the BBA particles from Oregon fire was higher than Creek fire. PLDR is a parameter reflecting the particle shape, i.e. spherical particles have

PLDR of zero while morphologically complicated particles such as dust and ice crystals have high PLDRs. $PLDR_{532}$ is about 0.02–0.05 in Creek plume after 3–7 days' transport in the upper troposphere. In comparison, the $PLDR_{532}$ in Oregon plume is about 0.10–0.15 after 1–7 days' transport. The difference indicates that BBA particles generated from the Creek fire and Oregon fire may have some differences in the morphology and/or other properties.





## 2.2 Sun/sky photometer observations near wildfire source

NASA_Ames and PNNL are two AERONET sites located near the fire activities, as shown in Figure 1(b). PNNL was mainly impacted by the Oregon smoke during 12–19 September. The $AOD_{500}$ (Aerosol Optical Depth at 500 nm) was about 0.2 and $AE_{340-500}$ (Angström exponent between 340 and 500 nm) about 1.5 in 01–08 September, before the arrival of the Oregon smoke. When the Oregon plumes arrived at PNNL on 12 September, the AOD sharply increased and the AE increased at the same time. The $AOD_{500}$ significantly increased during the period 12-19 September and the maximum $AOD_{500}$ of 4.0 (with

$AE_{340-500}$=0.2). NASA_Ames is in the west of the Creek fire and was influenced firstly by the Creek fire and then by the Oregon fire. $AOD_{500}$ on 04–08 September when Creek smoke particles dominated was less than 1.0 (with $AE_{340-500}$ ∼1.2), because the center of the Creek plumes propagated eastwardly and only a part of it drifted to the west. The Oregon plumes started on 09 September, as indicated by a sharp increase of AOD and a decrease of AE. The peak $AOD_{500}$ of 5.8, corresponding to $AE_{340-500}$ ∼0.0, was detected on 12 September in NASA_Ames. Similar to the observation at PNNL, $AE_{340-500}$ obviously

decreased during the episodes of Oregon smoke. The decrease of $AE_{340-500}$ indicated the increase of particle size in the Oregon BBA plumes, which is in agreement with the CALIPSO observations.

## 2.3 Lidar and photometer observations in Lille, France

The lidar involved in this study is a multi-wavelength Mie-Raman-polarization-fluorescence lidar, LILAS, operated at Laboratoire d'optique atmosphérique (LOA), University of Lille, France. The lidar system uses a Nd: YAG laser and has three emitting

wavelengths, i.e. 355, 532 and 1064 nm. The backscattered lidar signals are collected at three elastic wavelengths each coupled with a parallel channel and a cross channel, and three Raman channels at 387 (vibration Raman of nitrogen), 530 (rotational Raman of nitrogen) and 408 nm (water vapor channel). Since December 2019, the water vapor channel was replaced by a fluorescence channel centered at 466 nm (excitation wavelength: 355 nm) in order to profile atmospheric fluorescence. A full description of this configuration and the results of the feasibility test can be found in Veselovskii et al. (2020). With this con-

figuration, we can derive the height-resolved dataset $2\alpha + 3\beta + 3\delta + \phi$ ($\alpha$: extinction coefficient, $\beta$: backscattering coefficient, $\delta$: particle depolarization ratio, $\phi$: fluorescence capacity). The methodology has been presented in our previous publications (Hu et al., 2019; Veselovskii et al., 2020, 2021) and will not be repeated in this paper. The fluorescence signal is attributed to certain molecules that could absorb in the incident laser light and re-emit it at longer wavelength. The fluorescence capacity represents the capacity of aerosol particles in producing fluorescence, hence it is linked to particle chemical compositions.

Figure 4 shows lidar observations in the period 11-22 September 2020. Figure 4(a) and 4(b) show the range corrected lidar signal and the volume depolarization ratio at 1064 nm, respectively. The background aerosols in Lille are usually low depolarizing fine particles in the boundary layer. During the observational period, the BBA layers distributing at 2000-14000 m and polluted dust in the boundary layer were the two aerosol species. The smoke plumes are obvious in Figure 4(a) and disappear in Figure 4(b), because BBA particles have very weak depolarization at 1064 nm. The most intense smoke plumes are observed

on 11-12 September, corresponding to $AOD_{500} \approx 0.65$ and $AE_{340-500} \approx 0.70$, shown in Figure 4(c). Polluted dust from west Africa arrived in Lille on 14 September and concentrated mainly in the boundary layer. The intrusion of polluted dust does





not cause a significant decrease of $AE_{340-500}$, which can be explained by mixing with a non-negligible fraction of fine mode aerosol component, such as industrial pollution and smoke. On 17-18 September, a decrease of $AE_{340-500}$ is detected when a smoke plume appeared at 6000-10000 m. The $AOD_{500}$ and $AE_{340-500}$ on 17 September is about 0.22±0.02 and 0.0±0.20,
respectively.

Figure 5 plots the intensive parameters in the BBA layers detected in time intervals in September 2020. The observation time and the height range of the layers are resented in Table 1. Lidar ratio, i.e. the ratio between extinction coefficient and backscatter coefficient, is correlated with multiple factors, such as the particle shape, size and refractive indices. $LR_{355}$ being lower than $LR_{532}$ is a typical feature of aged smoke particles from North America and Siberian wildfires. The values of lidar ratios
varied in the range of 20–50 sr at 355 nm and 42-90 sr at 532 nm. The spectral dependence of the PLDRs is in agreement with the aged wildfire smoke particles in the literature. The variation ranges of PLDRs are respectively 0.03–0.22, 0.02–0.16 and 0.01–0.03 at 355, 532 and 1064 nm in the investigated cases. The PLDRs on 18 and 19 September (P7 and P8) are higher than other days, with mean PLDR equals to 0.16 at 355 nm, 0.12 at 532 nm and 0.02 at 1064 nm. The fluorescence capacity varied from $1.0 \times 10^{-4}$ to $4.0 \times 10^{-4}$. The $BAE_{355-532}$ (Angström exponent related to backscatter coefficient) is in the range
of 1.5–2.5. Whereas, the variation of $EAE_{355-532}$ (Angström exponent related to extinction coefficient) is stronger. Before 17 September, the $EAE_{355-532}$ varied in 0.5-1.5 and after this day, it dropped to below zero. Apart from the temporal variations, vertical variations in the BBA layers are also significant, such as lidar ratios in on 12 and 14 September, PLDRs on 11 and 18 September. Such variations are possibly indications of the variabilities in the burning materials, combustion conditions and aging process. Two events corresponding to Creek fire and Oregon fire, respectively, are selected and analyzed in detail in the
next section.

## 3  Case study

### 3.1  Case 1: 11-12 September 2020

The BBA plumes detected in the night of 11-12 September 2020 are attributed to the Creek fire and have traveled about 5-7
days. Figure 6(a) shows the 4-day back trajectories overlaid with UVAI map on 8 September 2020. Two areas with intense UVAI are indicated on the map, one over the western coast of the US and the other one over the Great Lakes. The former was mostly generated by the Oregon fire, which started on 7 September. The latter was emitted by the Creek fire on 4-6 September and transported to Europe via an 'expressway'. Figure 7 shows the range corrected lidar signal (RCS) and volume linear depolarization ratio (VLDR) and the relative fluorescence signal, $\frac{P_{466}(z)}{P_{387}(z)}$, on 11-12 September. The fluorescence observations
are only available in nighttime when there is no sunlight interference. The BBA layer mainly distributed at 5000-10000 m and was characterized by strong fluorescence. Ice particles with strong depolarization were detected within the smoke layers above 8000 m.

Figure 8 plots the parameters obtained from averaged observations between 22:00 UTC, 11 September and 03:00 UTC, 12 September. The extinction coefficients peaked at about 6200 m, with about 180 $Mm^{-1}$ at 355 nm and 140 $Mm^{-1}$ at 532 nm.





The $EAE_{355-532}$ was about 0.5 and the $BAE_{355-532}$ is about 2.0. The $EAE_{355-532}$ is lower than the $AE_{340-500}$ measured at NASA_Ames when influenced by the Creek fire plumes, indicating the particle growth during the aging process. The lidar ratios, which are 40±6 sr at 355 and 70±11 sr at 532 nm, are typical values of aged smoke particles (Haarig et al., 2018; Hu et al., 2019; Ohneiser et al., 2020; Peterson et al., 2018). The PLDRs show strong vertical variations through the BBA layer. In the core of the smoke, i.e. 5000-6000 m, the PLDRs at the three wavelengths are all below 0.03. In the range of 6500-9000

m (the plume edge), the PLDRs at 355 nm and 532 nm increased significantly, with about 0.15 at 355 nm and 0.12 at 532 nm, indicating the change of particle morphology versus altitude. $PLDR_{1064}$ shows no vertical variations. A slight decreasing trend is observed in the $EAE_{355-532}$ when the height increases, which indicate the increase of particle size. The increasing trend of $BAE_{355-532}$ is stronger than $EAE_{355-532}$. The layer at 5000-6000 m is also characterized with slightly stronger fluorescence capacity ($\sim 3.8\times10^{-4}$ $sr^{-1}m^{-1}$ ) and higher water vapor content compared to the smoke in the range of 6500-9000 m.

According to laboratory studies, the main fluorophores in smoke are the humic-like substance and poly-aromatic hydrocarbons Garra et al. (2015); Zhang et al. (2019), which are common products of the combustion of biomass. During the aging process, the formation of some SOA could also contribute to the fluorescence signal (Lee et al., 2013). Therefore, temporal and spatial variations of fluorescence capacity of smoke are expectable during the transport, and such variation is dynamic and controlled factors, such as oxidation and evaporation of some chemical species.

Figure 8(g) presents the WVMR, RH and temperature profiles measured by radio sounding at Beauvechain, Belgium station, 120 km to the lidar station in Lille. The RH with respect to ice ($RH_{ice}$) was calculated with the radio sounding measurements (Jarraud, 2008). The variation of the BBA layer is well reflected by the profile of WVMR, which is also a tracer of airmass. It indicates that the BBA plumes were also captured by the radio sounding measurements at Beauvechain due to the large spatial coverage of the plumes. The WVMR increases from the plume center to the edge, suggesting that the WVMR is an important

role in the aging process.

## 3.2    Case 2: 17-18 September 2020

The smoke layers at 5000-9000 m on 17-18 September 2020 are attributed to the Oregon fire. The plumes underwent 7-12 days of transport before arriving at the observation site. Figure 6(b) presents the 6-day back trajectories overlaid with UVAI from OMPS on 14 September 2020. The plumes covered large areas of America and spread to the eastern Pacific Ocean. The lidar

quicklooks on 17-18 September are shown in Figure 9.

Profiles obtained from averaged measurements between 22:30 UTC, 17 September and 03:00 UTC, 18 September are plotted in Figure 10. The extinction coefficients peaked at about 7500 m, with 80-85±9 $Mm^{-1}$ at both 355 and 532 nm. The $EAE_{355-532}$ is slightly negative, -0.30±0.30, lower than in Case 1. This feature indicates that smoke particles in Case 2 are bigger than those in Case 1. The $BAE_{355-532}$ is comparable with that in Case 1. Compared to Case 1, lidar ratios in Case 2 are lower,

with 24±4 sr at 355 nm and 50±8 sr at 532 nm. These values are lower than many reported lidar ratios of aged smoke (Haarig et al., 2018; Hu et al., 2019; Ohneiser et al., 2020). At the 355 nm, the measured lidar ratio is close to that of cirrus clouds (Haarig et al., 2016; Hu, 2018). Such low lidar ratios were detected by Ortiz-Amezcua et al. (2017) in Raman lidar observations of long-range transported North America smoke over two European lidar stations, Leipzig and Granada. The observations in



Ortiz-Amezcua et al. (2017) show mean $EAE_{355-532}$ of 0.2-0.3, mean lidar ratios of 23-25 sr at 355 nm and 47-51 sr at 532
nm. PLDRs in Case 2 are 0.15±0.03 to 0.18±0.04 at 355 nm, 0.12±0.02 to 0.14±0.03 at 532 nm and about 0.02 at 1064 nm,
higher than those in Case 1 at all wavelengths. The fluorescence capacity varied in 2.0-3.0×10$^{-4}$ at 5300-8000 m, which is
lower than in Case 1. The BBA plumes in Case 2 traveled longer time and were diluted to a higher extent. During this process,
the removal (e.g. dry and wet deposition), transformation (oxidation, dissociation) and evaporation of versatile species could
lead to the decrease of the main fluorophores. Whereas, the difference in vegetation type and combustion conditions between
Case 1 and 2 cannot be excluded.

A sharp increase of $PLDR_{1064}$ to nearly 0.10 was detected at 8600 m, indicating the presence of ice crystals. At such altitude, the
temperature was about -41°C to -34°C and the $RH_{ice}$ was 100-120%, according to model data (GDAS: global data assimilation
system) and radio sounding measurements. Lidar signals before and during the presence of cirrus have been averaged and
shown in Figure 11. The fluorescence backscattering coefficient in the cirrus layer is as high as before the presence of ice
crystals, indicating that the cirrus layer contains non-negligible BBA particles. The decrease of fluorescence capacity in cirrus
layer is mainly due to the enhancement of elastic scattering associated with the presence of ice crystals. This observation is an
indication of BBA particles acting as INP, although the possibility that the ice crystals formed elsewhere cannot be excluded.
Considering that there were almost no high clouds, in liquid or solid phase, over Lille on 17-18 September, these ice crystals
were very likely formed through deposition freezing mode or pore condensation and freezing mode (Cziczo et al., 2013; Umo
et al., 2019).

## 4    Discussion

### 4.1    Optical properties: AE, PLDRs, lidar ratios and fluorescence capacity

In this study, we demonstrate that the key intensive BBA optical properties, i.e. the Ansgtröm exponent ($AE_{340-500}$/$EAE_{355-532}$),
fluorescence capacity, lidar ratios and PLDRs, have obvious vertical and temporal variations. These variations are potentially
linked to complex factors, such as the fuel types, combustion conditions and aging process. Angström exponent, $AE$/$EAE_{355-532}$,
decreased when the plumes transported from the source region to the observation site. This decrease reflects the increase of
particles size during the transport, as has been observed by both in-situ and remote sensing studies. The increase of particle size
in long-range transport is the result of condensational growth and coagulation of BBA particles (Müller et al., 2007; Haarig
et al., 2018; Garofalo et al., 2019; Hodshire et al., 2019).

The spectral dependence of PLDRs and lidar ratios is also consistent with previous observations of aged North American BBA
(Burton et al., 2015; Haarig et al., 2018; Hu et al., 2019). In this study, the PLDRs demonstrate a strong dependence on altitude,
i.e. the higher altitude the higher PLDRs. The vertical variation of PLDRs has been discussed in previous studies. During the
episode of Canadian smoke over Europe, lidar systems in Leipzig and Lille both found that the tropospheric BBA plumes si-
multaneously detected with the UTLS plume had much lower PLDRs, smaller than 0.10 at all wavelengths. Baars et al. (2019)
summarized the Canadian smoke observations from different lidar stations within EARLINET and concluded that the PLDRs
tend to decrease with travel time. Baars et al. (2019) explained that aged BBAs are mostly BC-containing and OC-coated



aggregates and the growth of coating during aging process makes the particles more spherical. However, this argument is in conflict with the observations of Ohneiser et al. (2020), who found that in the 2020 Australian wildfires smoke layers above 18 km mostly have PLDR (at 532 nm) greater than 0.15, obviously higher than those below this altitude in the stratosphere.

Given that it takes time for the absorbing smoke plumes to raise to higher altitude by self-lifting, which is claimed by Ohneiser et al. (2020) and other researchers (Laat et al., 2012; Hu et al., 2019; Torres et al., 2020), plumes at higher altitude tend to be older. Yu et al. (2019) mentioned that the significant PLDRs should not be simply explained by the presence of fractal BC with non-spherical OC coating and hypothesized that the OC-coated particles should be solid because they freeze or effloresce at stratospheric temperatures. This argument can explain why the PLDRs of BBAs in the stratosphere are higher than those in the

troposphere. However, it cannot fully explain the sharp vertical variation of PLDRs from the core to the edge of the same BBA plume, as indicated in Case 1 of this study. Such variation indicates that the aging process is inhomogeneous in the plumes and may be sensitive to humidity, temperature and aerosol concentration. Hodshire et al. (2021) found that on the plume edges, the organic aerosol is more oxygenated compared to the plume cores. Studies demonstrated that the evaporation of condensed liquid water on the surface of BBA contributes strongly to the collapse of fractal-like aggregates into compact particles (Kütz

and Schmidt-Ott, 1992; Ma et al., 2013). The organic aerosols, the main contributor of the BBA mass, can exist in amorphous state, which means that they can transform into liquid, semi-solid and solid (glassy state) in response to temperatures and humidities (Koop et al., 2011; Berkemeier et al., 2014; Knopf et al., 2018). This special property influences not only the light scattering behavior of BBA particles but also their abilities of ice nucleation. However, the current theory cannot explain the observed variability in the evolution of BBA properties.

The vertical variation of lidar ratios are also obvious in September (Figure 5), however, such variations are resulted not only from the variability of BBA properties, but also the artifacts in the calculation of the extinction-to-backscatter ratio. The calculation of extinction coefficient requires vertical smoothing, while the backscatter coefficient does not. Hence, the division of the two values could lead to some fake vertical fluctuations, especially in the range where aerosol concentration changes fast. From this aspect, the fluorescence capacity is a better parameter for aerosol typing especially in low aerosol concentrations.

The variation of lidar ratios, also the PLDRs and $EAE_{355-532}$ and fluorescence capacity, from the Creek fire to the Oregon fire is a clear indication of the variability of BBA properties originated from different fires activities. The decrease of lidar ratios in the Oregon plumes is probably due to the decrease of aerosol absorption, according to the modeling of aerosol optical properties. However, other possible reasons such as particle morphology and hygroscopic effects can also impact the lidar ratio and cannot be excluded. The varied fluorescence capacity may reflect, at least to some extent, the changes of aerosol absorption,

because aerosols absorb the incident radiation and re-emit the fluorescence. More observations and collocated measurements are required to explore the links between the fluorescence capacity and aerosol absorption.

Many efforts have been done to reproduce the light scattering properties of aged BBAs, such as absorption, PLDRs and lidar ratios, using different aerosol models. Liu and Mishchenko (2020) and Gialitaki et al. (2020) have successfully modeled the spectral PLDRs and lidar ratios observed in aged Canadian BBA plumes using near spherical spheroidal and Chebyshev parti-

cles. Liu and Mishchenko (2018) used 11 different morphology models, ranging from bare soot to completely embedded soot mixtures, and simulated several scattering parameters, such as the absorption enhancement, absorption Angström exponent





and PLDRs. Liu and Mishchenko (2018) found that the absorption enhancement is affected by particle size, morphology and the thickness of coating. They also reproduced the spectral PLDRs observed by Burton et al. (2018) and concluded that multi-wavelength PLDR measurements can be used to identify the presence of morphologically complex carbonaceous particles.

The results of modeling demonstrate that composition and morphology of aged BBA are essential parameters that determine their properties. However, these studies are currently piecewise and hardly comparable, because they used different scattering models. To achieve a better characterization, more measurements and new information are needed. The idea of including laser-induced fluorescence is to extend the optical dataset and refine aerosol characterization, because with fluorescence information we could get a closer insight about the particle chemical compositions.


### 4.2    BBA acting as INP

Handful studies have shown that BBAs may serve as a source of INP (Petters et al., 2009; McCluskey et al., 2014; Prenni et al., 2012), but existing results of ice nucleation onset conditions are highly varied. The determination of the ice-active components in BBA and their mechanisms of ice nucleation are inadequate. Previous studies on ice nucleating abilities of BBA are mostly

about soot particles, OC and inorganic components. Recent studies based on experimental data and modeling claimed that BC is unimportant in ice cloud formation (Vergara-Temprado et al., 2018; Schill et al., 2020). The role the amorphous organic matter in the ice nucleating ability of BBA matter is uncertain. The phase state of organic matter influence the water diffusivity and the timescales of the phase transitions, which could modulate the pathways of ice formation (Mikhailov et al., 2009; Berkemeier et al., 2014; Wang et al., 2012; Lienhard et al., 2015; Shiraiwa et al., 2017; Price et al., 2015; Reid et al., 2018).

Laboratory studies on organic INPs are mostly limited to pure organics, such as sucrose and levoglucosan etc, while OA in ambient conditions have complex origins . As a results, the conditions for initiating ice nucleation required by different organics are highly varying in literature data, which blocked us from clarifying the role of organic aerosols in ice formation (Kanji et al., 2017). Recently Jahn et al. (2020) pointed out the importance of inorganic elements, which naturally existed in biomass and can be transformed into potentially ice-active minerals. Jahn et al. (2020) presented that these particles containing

ice-active minerals are the major INPs in BBA because they could nucleate ice in a wide range of temperatures, up to -13°C. The influence of BBA aging on its ability of INP is also an important issue to be taken into account. Several studies have reported the enhancement of ice-nucleating ability of BBA after aging process, although they have not yet reached a consensus on the mechanisms (Brooks et al., 2014; Lupi and Molinero, 2014; Mahrt et al., 2020; Jahl et al., 2021). The establishment of a thorough determination the mechanism of ice nucleation by BBA particles requires more data from field measurements and

laboratory measurements based on advance aerosol particle analytical techniques.

## 5    Conclusions

This study reported the observations of long-range transported BBA plumes originated from wildfires in West America in September 2020. The analysis involves data from multiple sources, including satellite measurements, ground-based sun/sky



photometer and lidar and models. In September 2020, two intense fires – the Creek fire and Oregon fire, have been identified by

OMPS and CALIPSO observations. After 1-2 weeks transport, these BBA plumes in the troposphere were detected by LILAS, a multi-wavelength Mie-Raman polarization fluorescence lidar in Lille, north of France. The optical properties $2\alpha+3\beta+3\delta+\phi$ obtained from lidar measurements provided a good dataset for BBA particle characterization. Our results demonstrated typical features of aged BBA particles that have been reported in previous research, i.e. decreased Angström exponent, enhanced PL-DRs at shorter wavelengths and lidar ratio at 532 nm greater than at 355 nm. These characteristics are very likely resulted from

the growth of particle size and changes of particle morphology during the aging process. Moreover, our results showed strong variations of particle intensive parameters dependent on altitudes and combustion events. In the Creek plume, PLDRs increased with altitudes and from the core ($\text{PLDR}_{355,532,1064} < 0.03$) to the edge of the plume ($\text{PLDR}_{355,532,1064} \approx 0.15, 0.12, 0.01$). Such characteristics suggest that the aging process is not homogeneous in the BBA plume and may dependent on environmental conditions such as humidities, particle concentration, temperature and so on. Compared to the Creek plume, the decrease

of lidar ratios, $\text{EAE}_{355-532}$ (and AE) and fluorescence capacity $\phi$ in Oregon plumes, as well as the enhancement of $\text{PLDR}_{532}$ pointed out the variability of BBA properties in different combustion events. The low lidar ratios in the Oregon plumes are possible linked to the presence of low absorbing particles. The fluorescence capacity derived from the fluorescence channel at 466 nm turned to be a better parameter than lidar ratio in the identification of smoke BBAs, because of its high sensitivity to low concentrations. Benefitted from the fluorescence channel, lidar measurements allow us to directly detect the signature

of BBA particles in ice clouds. During the smoke episode, BBA particles existing in cirrus cloud layers have been frequently detected. Our results suggest that aged BBA particles could nucleate ice crystals under atmospheric conditions in the UTLS. The fluorescence capacity of future lidar LIFE (Laser-induced fluorescence explorer) at LOA will be enhanced with more powerful laser and more fluorescence channels in order to obtain higher quality observations. As wildfire emissions are becoming an important atmospheric component, our observations could help improve the BBA characterization and lead to a

better understanding of the BBA aging process and the role of BBA in ice nucleation.

*Data availability.* The data is available upon any request.

*Author contributions.* QH performed lidar measurements, developped the code for data analysis and wrote the manuscript. PG is the PI of the project, supervised the experiments and improved the manuscript. IV helped in data analysis and establishment of the scientific arguments. TP helped perform lidar measurements and is in charge of the maintenance and operation of lidar system.

*Competing interests.* The authors declare no competing interests.



*Acknowledgements.*  We acknowledge the CaPPA project (Chemical and Physical Properties of the Atmosphere), funded by the ANR (French National Research Agency) through the PIA (Programme d'Investissement d'Avenir) under contract « ANR-11-LABX-0005-01 », the Regional Council « Hauts-de-France » and the FEDER (European Funds for Regional Economic Development). We also would like to thank the following organizations and individuals for the financial support and for sharing their data and expertise: Service National d'Observation PHOTONS/AERONET-EARLINET, ESA/QA4EO, AERONET (PIs of the sites involved in this study: Roy Johnson, Brent Holben, Philippe Goloub), CPER CLIMIBIO, Russian Science Foundation (project 21-17-00114) and Lab Agora.



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





**Table 1.** BBA layers detected by LILAS system in Lille

| NO of period | Time interval [UTC] | Height [m] |
|:---:|:---:|:---:|
| P1 | 2020-09-11, 02:00:00 – 2020-09-11, 04:00:00 | 5500 – 7500 |
| P2 | 2020-09-11, 22:00:00 – 2020-09-12, 02:00:00 | 5500 – 8500 |
| P3 | 2020-09-12, 20:00:00 – 2020-09-12, 23:00:00 | 2500 – 3800 |
| P4 | 2020-09-14, 00:00:00 – 2020-09-14, 03:00:00 | 5500 – 7500 |
| P5 | 2020-09-14, 22:00:00 – 2020-09-15, 02:00:00 | 5500 – 6500 |
| P6 | 2020-09-15, 02:00:00 – 2020-09-15, 04:00:00 | 4500 – 7200 |
| P7 | 2020-09-17, 22:00:00 – 2020-09-18, 03:00:00 | 5400 – 8500 |
| P8 | 2020-09-18, 20:00:00 – 2020-09-18, 22:00:00 | 4800 – 6500 |
| P9 | 2020-09-20, 20:00:00 – 2020-09-21, 00:00:00 | 4400 – 6200 |



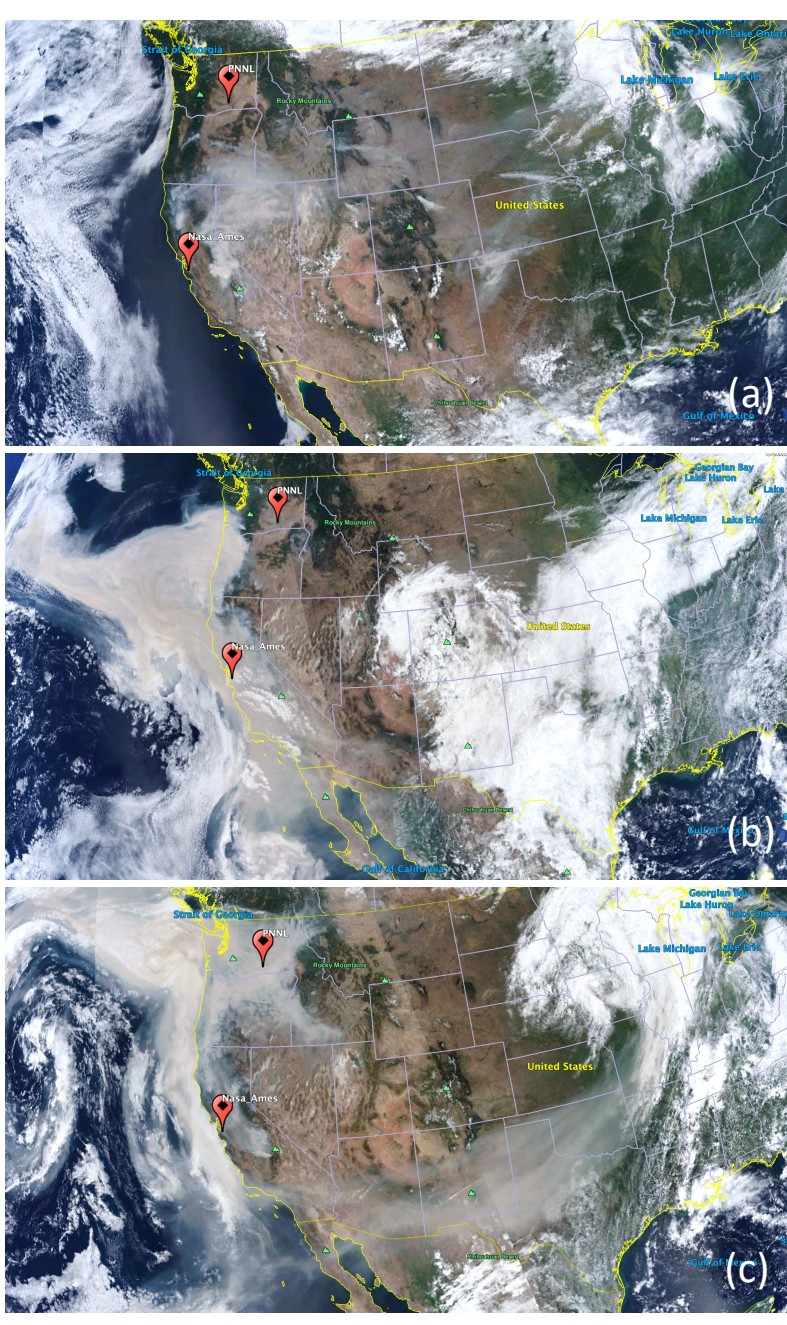

**Figure 1.** True color images from OMPS onboard Suomi NPP. (a) 06 September 2020, (b) 10 September 2020, (c) 12 September 2020. The two AERONET observation sites: NASA_Ames (37.420N, 122W) and PNNL (46N, 119W). ©Google Maps



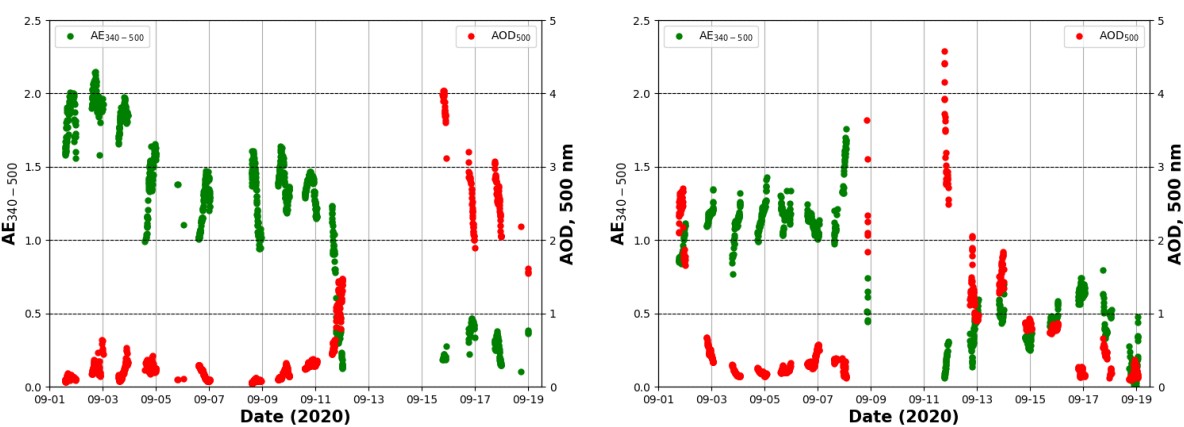

**Figure 2.** AOD and AE (Level 1.5) measured by AERONET sun/photometers at two observation sites: (a) PNNL (46N, 119W) and (b) NASA_Ames (37.420N, 122W). NASA_Ames is within the influence of both the Creek fire and the Oregon fire. PNNL is mainly impacted by the smoke of Oregon fire. The smoke of Oregon fire arrived at NASA_Ames on about 9 September and at PNNL on 12 September 2020. The arrival of the Oregon smoke caused significant increase of AOD and decease of the AE at both sites.

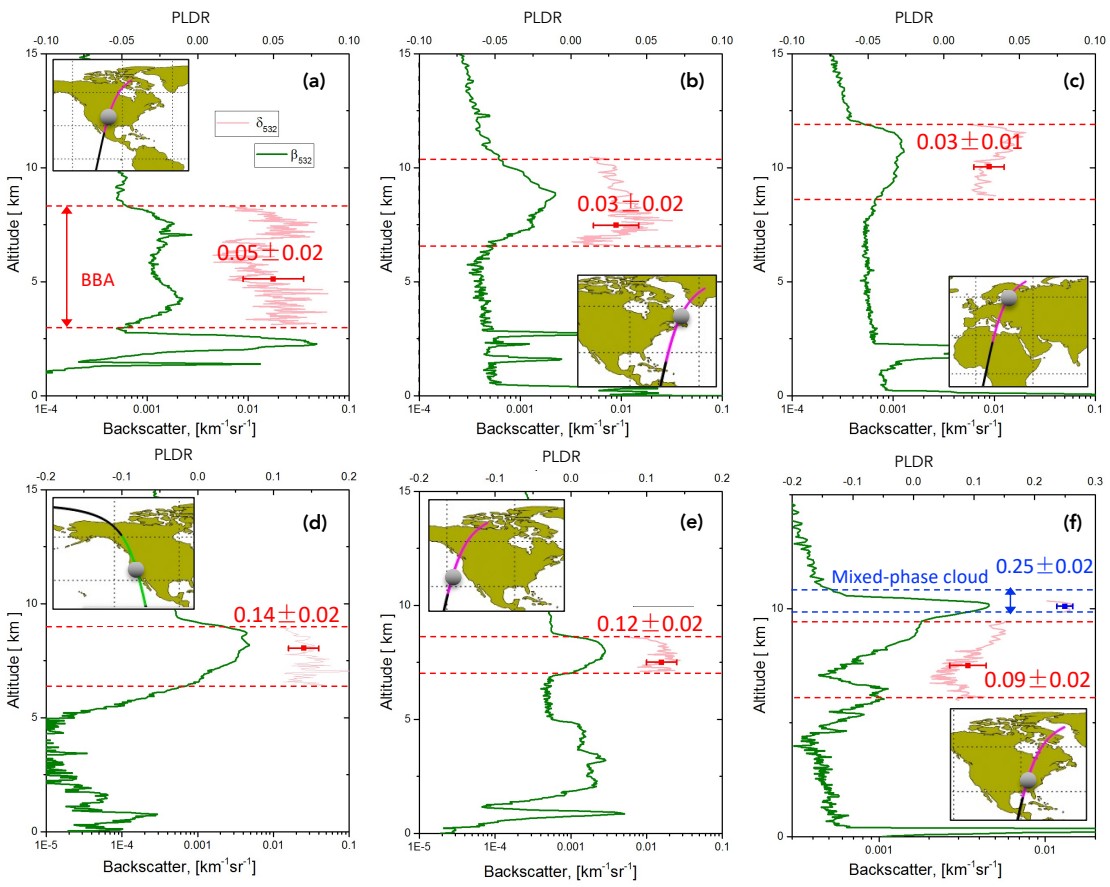

**Figure 3.** The backscattering coefficients and PLDR ratio at 532 nm measured by CALIPSO. The smoke plumes in (a), (b) and (c) are attributed to Creek fires on 05-07 September 2020, and in (d), (e) and (f) are attributed to Oregon fires. The inserted maps display the granules of CALIPSO and the locations of the grey dots represent the region where the measurements are averaged. (a) Central US, 2020-09-08. (b) East of Quebec, 2020-09-09. (c) Eastern Europe, 2020-09-11. (d) Western coast of US, 2020-09-10. (e) Western coast of US, 2020-09-11. (f) Eastern US, 2020-09-14.





**Figure 4.** Lidar and sun/sky photometer observations during the smoke episodes on 10-22 September 2020, Lille, France. (a) The range-corrected lidar signal at 1064 nm. (b) The volume depolarization ratio at 1064 nm. (c) The AOD and AE measured by the sun/sky photometer operated in Lille. The layers with strong depolarization appearing in the boundary layer on 14-16 September are polluted dust from Saharan region. The smoke layers from Californian smoke are mostly distributed in the free troposphere.

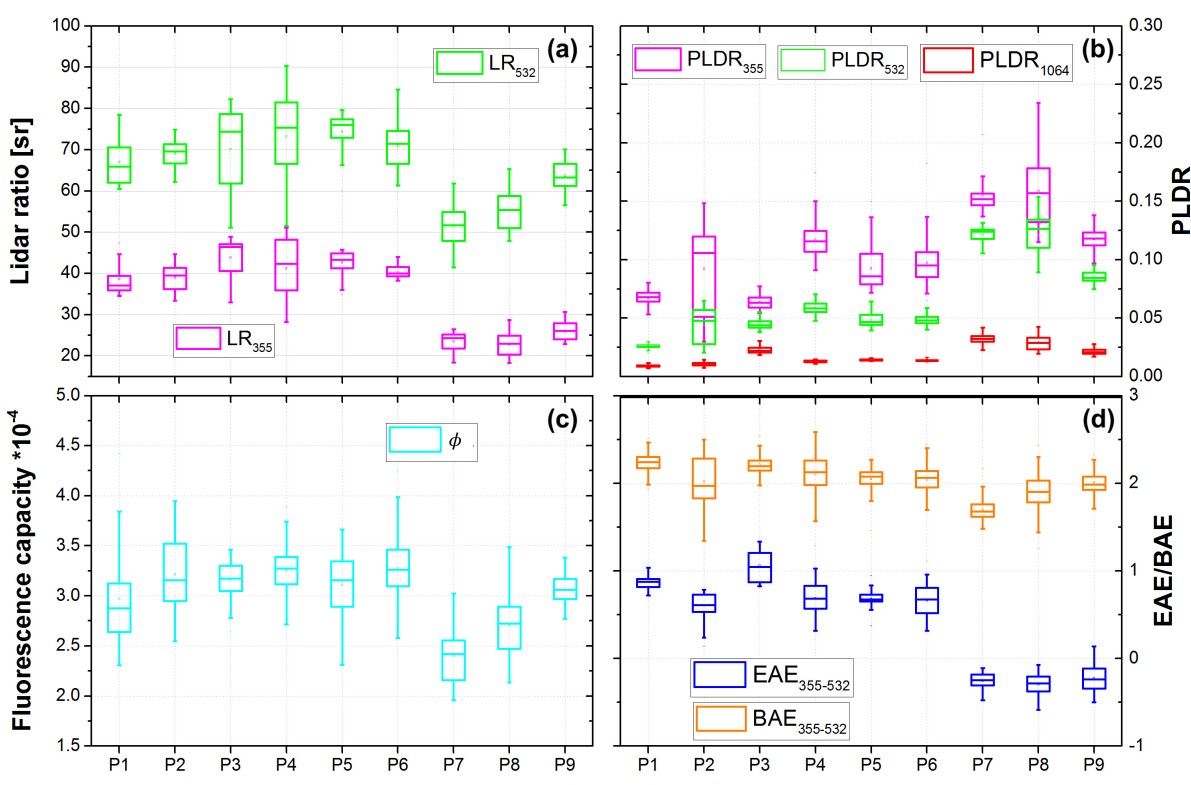

**Figure 5.** Box plots of the variabilities of BBA intensive parameters during the smoke episode in September 2020. The lower and upper edges of the box represents respectively the first and third quartiles of data. The lower and upper range-bars represent respectively the minimum and maximum. The dot and bar in the box indicate the mean value and the middle value, respectively. (a) Lidar ratios, (b) PLDRs, (c) fluorescence capacity, (d) $EAE_{355-532}$ and $BAE_{355-532}$.





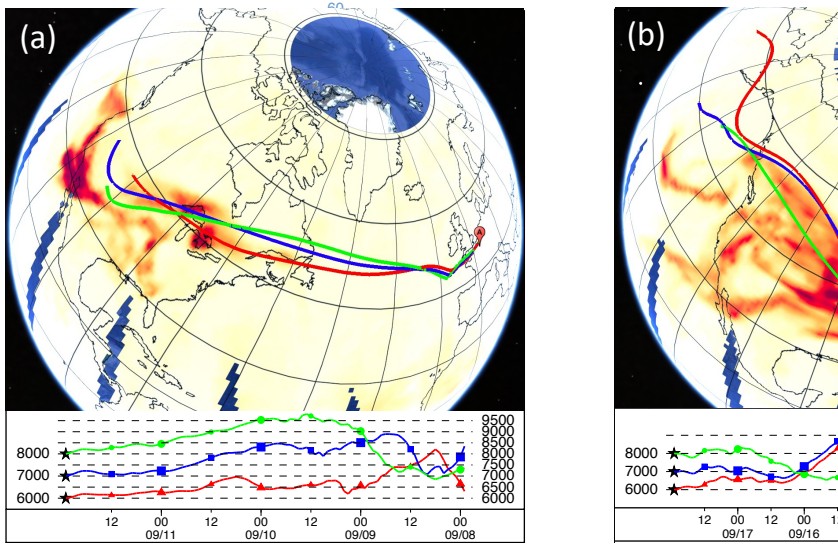

**Figure 6.** HYSPLIT back trajectories overlaid on the UVAI measured by OMPS onboard SUOMI-NPP satellite. (a) 96-hour back trajectories ending at 00:00 UTC, 12 September 2020 overlaid with UVAI on 08 September. (b)144-hour back trajectories ending at 00:00 UTC, 18 September 2020 overlaid with UVAI on 14 September. ©Google Earth.

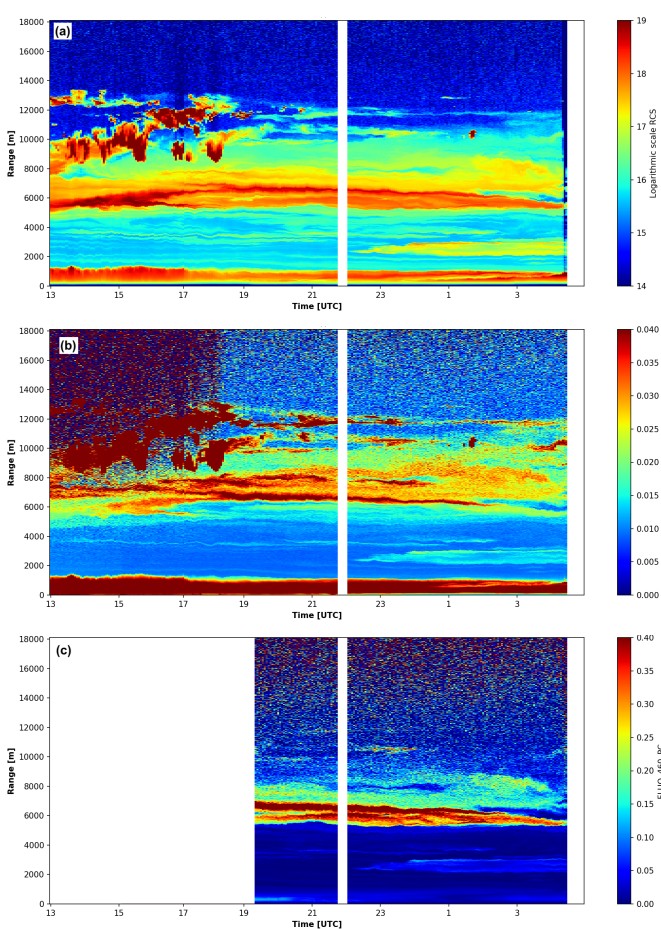

**Figure 7.** Lidar observation on 11-12 September 2020. (a) Range corrected lidar signal at 1064 nm, (b) volume depolarization ratio at 532 nm (c) backscatter coefficient of fluorescence at 466 nm. The white zone in the figures represents missing data due to the turnoff of the detectors (ex. to avoid daytime sunlight interferences) or intermediate operations during the measurements.



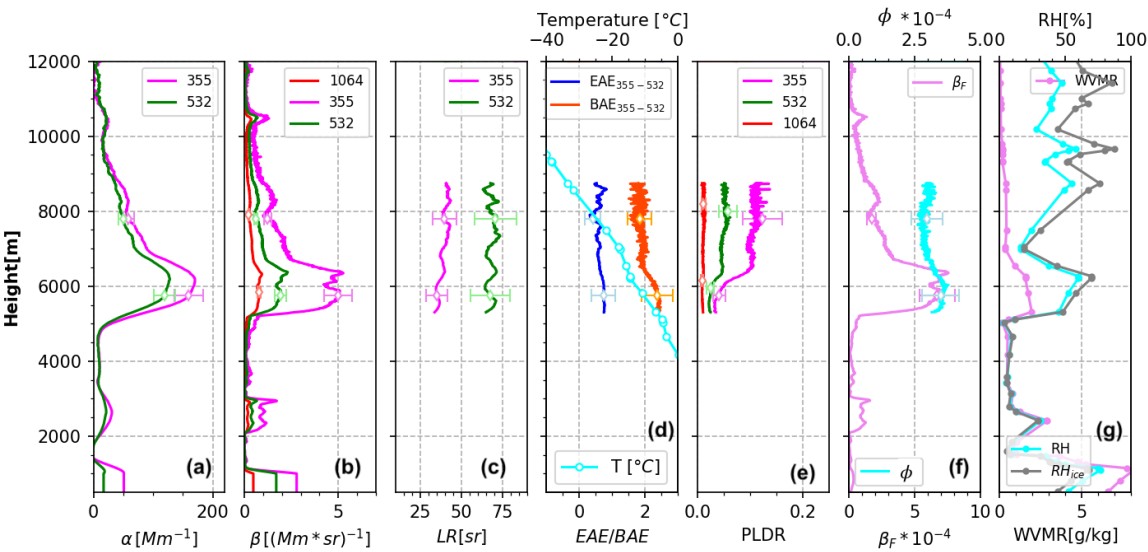

**Figure 8.** Aerosol vertical profiles from averaged measurements between 22:00, 11 September 2020 and 02:00, 12 September 2020,UTC. The extinction and backscatter coefficients at 355 and 532 nm were calculated using Raman technique (Ansmann et al., 1992) and the backscatter coefficient at 1064 nm was calculated by Klett method with an assumption of LR=50 sr. The radio sounding station is in Beauvecchain, Belgium, about 100 km from the lidar observation site. The error bars in the figure represent the statistical errors. The method of error estimation is presented in the Appendix of (Hu et al., 2019). The relative humidity to ice is calculated using the improved Magnus formula in Alduchov and Eskridge (1996).

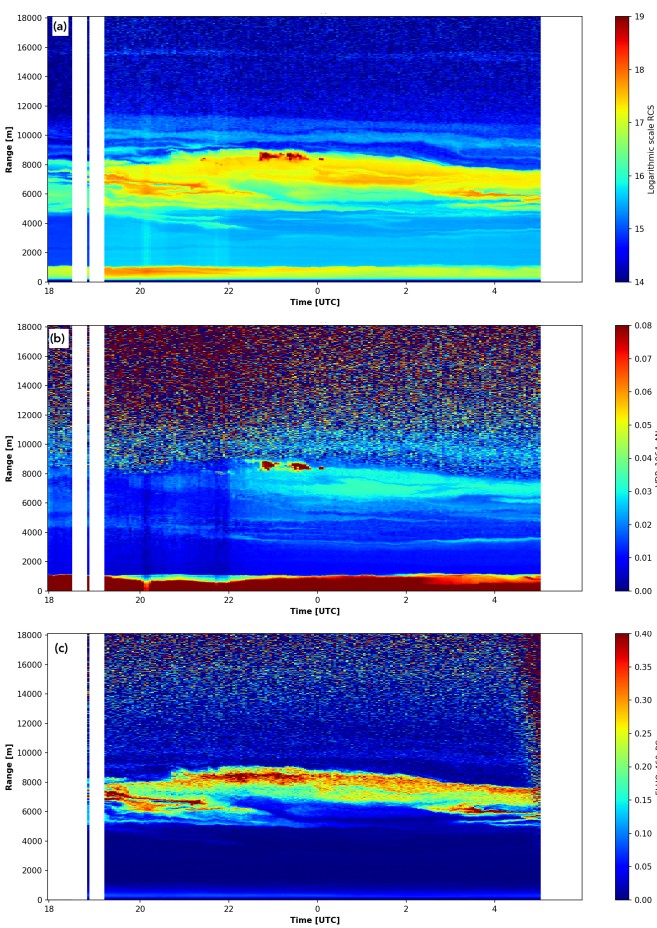

**Figure 9.** Lidar observation on 17-18 September 2020. (a) Range corrected lidar signal at 1064 nm, (b) volume depolarization ratio at 1064 nm, (c) backscatter coefficient of fluorescence at 466 nm.





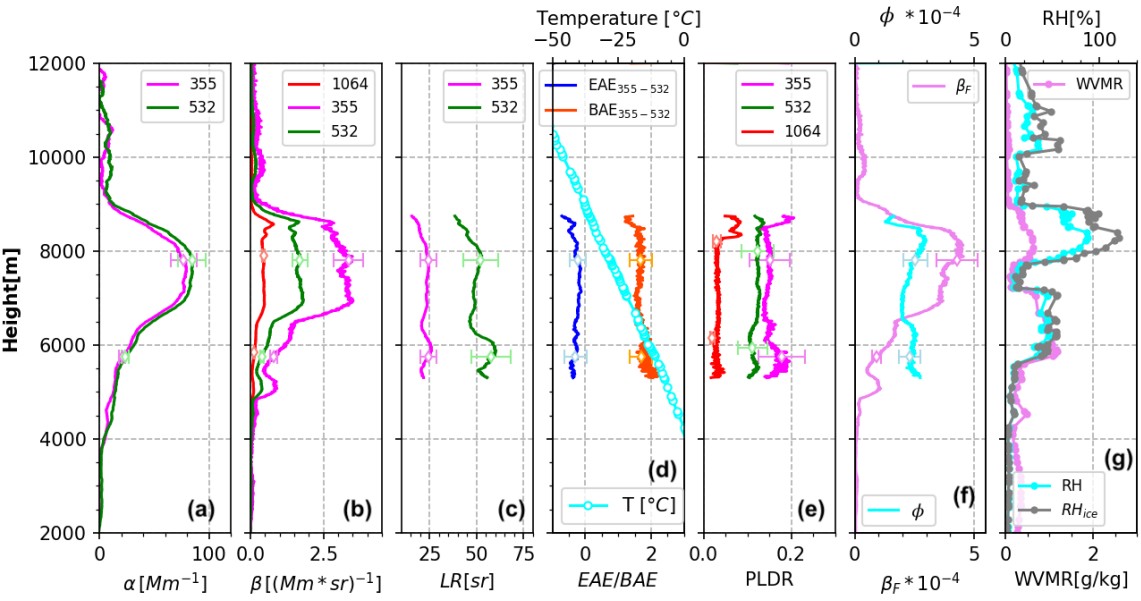

**Figure 10.** Aerosol vertical profiles from averaged measurements between 22:30, 17 September 2020 and 03:00, 18 September 2020,UTC. The calculation is done with the same method as in Figure 8. The radio sounding measurements are from Hertzmonceux, England, which is about 200 km from the observation site. Radio sounding at Beauvecchain, Belgium is not available on this day. The error bars represent the statistical errors, the same as in Figure 8.



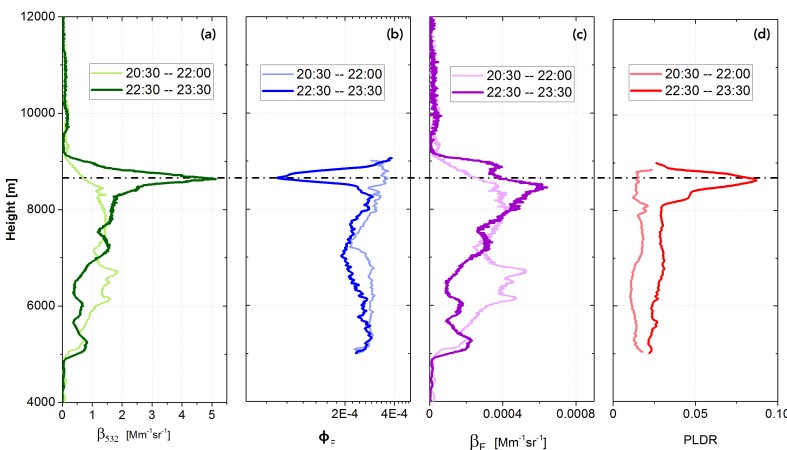

**Figure 11.** Averaged parameters in two consecutive periods: 20:30–22:30 UTC, 17 September 2020 (before the observation of ice crystals) and 22:30–23:30 UTC, 17 September 2020 (during ice crystal observation). The dash-dotted line represents the position of the ice crystals. (a) Backscattering coefficient at 532 nm, (b) fluorescence capacity at 466 nm, (c) fluorescence backscattering coefficient, (d) PLDR at 1064 nm.