# Peer review of "The characterization of long-range transported North American biomass burning plumes: what can a multi-wavelength Mie-Raman-polarization-fluorescence lidar provide?"

_Atmospheric Chemistry and Physics, 2021_

## Referee Comment (RC1)

**Review of manuscript ACP-2021-971**

The manuscript presents results from the observation of two events of aged biomass burning aerosol over Lille (northern France) using a lidar with 3 elastic channels (1064 nm, 532 nm, and 355 nm), 2 Raman channels (the $N_2$ vibro-rotational channel at 387 nm and a purely rotational channel at 530 nm), which provide the capacity to measure aerosol extinction coefficient – and lidar ratio – without assumptions, 3 depolarization channels (1064 nm, 532 nm, and 355 nm) to measure linear particle depolarization ratio, and 1 fluorescence channel at 466 nm serving to identify the biomass. Other satellite observations, from CALIPSO and from Suomi NPP, as well as ground-based observations from AERONET Sun/sky photometers, are used as ancillary data.

The results permit to identify different properties of the aerosol in two case studies, which can be related to different wildfire source areas and/or aging processes.

**General remarks**

The study carried out is interesting and points toward new methods to characterize aerosols (not only biomass burning), but it looks a little inconclusive. Probably, it would benefit of being less ambitious in its scope and focusing on the question of the title, "what can a multi-wavelength Mie-Raman-polarization-fluorescence lidar provide?", which is, in my view, a little lost in rather inconclusive discussions on optical properties and on the possibility of biomass burning aerosols acting as ice nucleating particles. Not that these issues should not be dealt with, but probably with less detail, getting more to the facts, i.e. to what the measurements show. Because lidars with Raman channels and depolarization capabilities have long existed, I suggest that the paper focus more on the additional information provided by the fluorescence-measurement capability of the lidar used in the study.

**Specific remarks**

1. Lines 141-142: the 530 nm Raman channel is described as "rotational Raman of nitrogen". However, it is unlikely that this channel picks up only lines from the nitrogen rotational Raman spectrum; most probably it also collects lines from the oxygen rotational Raman spectrum. Please check and correct if necessary.

2. Line 148: although for the lidar configuration and its capabilities the reader is referred to published references, I think that the less usual fluorescence capacity parameter would deserve an explicit definition in the text. Probably it would fit right after "fluorescence capacity" in line 148.

3. Line 184 and figure 7(c): there seems to be some undefinition as to what is represented in figure 7. The text (line 184) says that it shows the "relative fluorescence signal, $\frac{P_{466}(z)}{P_{387}(z)}$". First of all, in case it is this ratio what is represented, please specify if this ratio is calibrated. Second, the caption of figure 7 says that what is represented is the backscatter coefficient of fluorescence at 466 nm, which is not quite the ratio given in line 184. Check as well the caption of fig. 9.

4. Lines 185-186: strong depolarization seems to be unambiguously ascribed to the presence of ice particles ("Ice particles with strong depolarization were detected within the smoke layers above 8000 m"). My first remark is that a call to the figure showing that (figure 7(b)?) is missing; the period of time when the strong depolarization is observed should also be specified. Secondly, this strong depolarization seems to come from cirrus clouds. It is known that depolarization can also be produced by multiple scattering. Can multiple scattering be ruled out as the cause of depolarization in this case? What's the lidar field of view? Even if the presence of ice is plausible, I think that the evidence for ice being the cause of depolarization should be developed further.

5. Line 205: "Figure 8(g) presents the WVMR, RH and temperature profiles". However, figure 8(g) doesn't seem to contain a temperature profile.

6. Lines 209-210: Talking about figure 8(g), it is said that "The WVMR increases from the plume center to the edge, suggesting that the WVMR is an important role in the aging process." I don't see that increase in the WVMR profile in figure 8(g), rather that profile follows the profiles of $\beta$ and $\beta_F$.

7. Line 231: "A sharp increase of $PLDR_{1064}$ to nearly 0.10 was detected at 8600 m, indicating the presence of ice crystals". Two remarks: a) Where should the reader look for that sharp increase? b) See my second remark in point 4.

8. Line 275: "The vertical variation of lidar ratios are also obvious in September (Figure 5)". First, it's unnecessary to talk about September, as all the presented measurements were carried out in September. Second, I don't see that fig. 5(a) makes obvious that variation. Although a variation in the lidar ratios can be seen between periods P1-P6 and periods P7-P9, I don't see that this variation is related to some vertical variation. For example, in period P2 the BBA layer is detected between 5500 and 8500 m, and in period P7, with lower lidar ratio, between 5400 and 8500 m, which is sensibly the same range as for P2. Please check if the figure referred to should be figure 5 or else be more explicit about what the reader should pay attention to.

9. Line 279: "From this aspect, the fluorescence capacity is a better parameter for aerosol typing especially in low aerosol concentrations". Please be more specific as to the meaning of "typing" in this context.

**Formal remarks and typos**

1. In the bibliography, please insert the doi whenever possible.

2. As a general rule, I would prefer that the full wording of acronym is used before the acronym; for example, in line 68, I would prefer "showed enhanced particle linear depolarization ratios (PDLRs)" than the current "showed enhanced PLDRs (particle linear depolarization ratio)".

3. Line 31: "They could alternate the planetary radiation budget of the planet". I think that the meaning is "They could alter the radiation budget of the planet".

4. Lines 80-81: "could influence the ability in water diffusion". This sentence sounds strange. Should it be "could influence the ability for water diffusion", or there is a word missing? Please check.

5. Lines 96-97: ATOLL is defined (Atmospheric Observatory of LiLle), but is never used afterwards.

6. Figs. 1(c) and 2 are not cited in the text. Also, the text where the call to figure 2 should be inserted (section 2.2) comes after the call to figure 3 in section 2.1, so the numbering of these figures should be swapped.

7. Line 162: "and the height range of the layers are resented". "resented" should be "presented".

8. Line 162: please insert the acronym LR after the lidar ratio is introduced, e.g. "Lidar ratio (LR), i.e. the ratio between extinction coefficient and backscatter coefficient".

9. Lines 171-173: "Apart from the temporal variations, vertical variations in the BBA layers are also significant, such as lidar ratios in on 12 and 14 September, PLDRs on 11 and 18 September. Such variations are possibly indications of the variabilities in the burning materials, combustion conditions and aging process". Where are these lidar ratios and PLDRs represented? Where the reader should look at?

10. Lines 188-189: "Figure 8 plots the parameters obtained from averaged observations between 22:00 UTC, 11 September and 03:00 UTC, 12 September". However, the caption of figure 8 says that the averaging end time is 02:00 UTC.

11. Line 197: "The increasing trend". Should it be "decreasing"? (see the sentence just before and fig. 8).

12. Line 205: Although evident for the expert, the acronyms WVMR and RH have not been defined.

13. Line 253: "episode of Canadian smoke over Europe". Please insert references, even if it implies repeating some of those already given in line 251.

14. As general rule, increase the size of legends, labels and scales in the graphs. Be also explicit in the labels. For example, the label of the color bar of figure 4(b) should read "Volume depolarization ratio at 1064 nm" instead of "VDR_1064_AN". I don't know if specifying that measurements are obtained from analogue (AN) or photon counting (PC) channels is relevant in the context of this paper.

15. Figure 2: the labels (a) and (b) are missing in the figure panels.

16. Figure 5. Please explain that P1, P2,…P9 in the horizontal scales refer to the periods identified in table 1.

17. Figure 5 caption, 3rd line: "The dot and bar in the box indicate the mean value and the middle value". "middle value" should probably be "median value".

18. Although the English writing is very good in general, I would suggest a review by a native English speaker to fix some odd uses. Just as an example, in lines 12-13, the "varied" in the sentence "It reflects that the properties of aged BBA particles are highly varied" should probably be "variable"; in line 15 "than those" sounds better than "with those", etc.

---

## Author Comment (AC1)

**Review of manuscript ACP-2021-971**

The manuscript presents results from the observation of two events of aged biomass burning aerosol over Lille (northern France) using a lidar with 3 elastic channels (1064 nm, 532 nm, and 355 nm), 2 Raman channels (the $N_2$ vibro-rotational channel at 387 nm and a purely rotational channel at 530 nm), which provide the capacity to measure aerosol extinction coefficient – and lidar ratio – without assumptions, 3 depolarization channels (1064 nm, 532 nm, and 355 nm) to measure linear particle depolarization ratio, and 1 fluorescence channel at 466 nm serving to identify the biomass. Other satellite observations, from CALIPSO and from Suomi NPP, as well as ground-based observations from AERONET Sun/sky photometers, are used as ancillary data.

The results permit to identify different properties of the aerosol in two case studies, which can be related to different wildfire source areas and/or aging processes.

**General remarks**

The study carried out is interesting and points toward new methods to characterize aerosols (not only biomass burning), but it looks a little inconclusive. Probably, it would benefit of being less ambitious in its scope and focusing on the question of the title, "what can a multi-wavelength Mie-Raman-polarization-fluorescence lidar provide?", which is, in my view, a little lost in rather inconclusive discussions on optical properties and on the possibility of biomass burning aerosols acting as ice nucleating particles. Not that these issues should not be dealt with, but probably with less detail, getting more to the facts, i.e. to what the measurements show. Because lidars with Raman channels and depolarization capabilities have long existed, I suggest that the paper focus more on the additional information provided by the fluorescence-measurement capability of the lidar used in the study.

Reply: Thanks a lot for the suggestions. The changes of BBA properties during the aging process and BBA acting as ice nucleating particles are very interesting topics, which fluorescence measurements could contribute to. We agree that the discussion about the smoke aging and ice nucleation is a bit too long and inconclusive. We need more observations, which are not presented in this manuscript, to consolidate them. An article more relevant to ice nucleation in smoke layer has been accepted by ACP. Therefore, we decided to follow the suggestion of the review, i.e. shorten the discussion about ice nucleation and focus more on the fluorescence measurements. The modifications can be seen in the new version of manuscript.

**Specific remarks**

1. Lines 141-142: the 530 nm Raman channel is described as "rotational Raman of nitrogen". However, it is unlikely that this channel picks up only lines from the nitrogen rotational Raman spectrum; most probably it also collects lines from the oxygen rotational Raman spectrum. Please check and correct if necessary.

Reply: Yes, the 530 nm channel includes the rotational spectrum of both nitrogen and oxygen. The manuscript has been modified.

2. Line 148: although for the lidar configuration and its capabilities the reader is referred to published references, I think that the less usual fluorescence capacity parameter would deserve an explicit definition in the text. Probably it would fit right after "fluorescence capacity" in line 148.

Reply: Two equations are added to define fluorescence backscattering coefficient and fluorescence capacity. We also added some examples of fluorescence and non-fluorescence aerosols in the manuscript.

3. Line 184 and figure 7(c): there seems to be some undefinition as to what is represented in figure 7. The text (line 184) says that it shows the "relative fluorescence signal $\frac{P_{387}}{P_{466}}$, First of all, in case it is this ratio what is represented, please specify if this ratio is calibrated. Second, the caption of figure 7 says that what is represented is the backscatter coefficient of fluorescence at 466 nm, which is not quite the ratio given in line 184. Check as well the caption of fig. 9.

Change the caption.

4. Lines 185-186: strong depolarization seems to be unambiguously ascribed to the presence of ice particles ("Ice particles with strong depolarization were detected within the smoke layers above 8000 m"). My first remark is that a call to the figure showing that (figure 7(b)?) is missing; the period of time when the strong depolarization is observed should also be specified. Secondly, this strong depolarization seems to come from cirrus clouds. It is known that depolarization can also be produced by multiple scattering. Can multiple scattering be ruled out as the cause of depolarization in this case? What's the lidar field of view? Even if the presence of ice is plausible, I think that the evidence for ice being the cause of depolarization should be developed further.

Reply: (1) Reference to Figure 7(b) is added. (2) Indeed, it is not accurate to say that the increases of volume depolarization ratio at above 8000 m are all caused by the presence of ice particles, because some variations are aparently due to the fluctuation of BBA concentrations. To be more accurate, the sentence is changed to "Ice particle featured with increased depolarization ratios were detected in some high BBA layers, e.g at 8000-1000 m at 17:00-18:00 UTC ". Multiple scattering can be excluded for causing high depolarization ratios in the smoke layer, because firstly the lidar field of view is small, i.e. 1 mrad, and secondly in this case, the concentration of BBA was too low to produce so strong multiple  scattering effect.

5. Line 205: "Figure 8(g) presents the WVMR, RH and temperature profiles". However, figure 8(g) doesn't seem to contain a temperature profile.

Reply: It was a mistake. The temperature profile is plotted in Figure 8(d). The caption of the figure and the text are corrected now.

6. Lines 209-210: Talking about figure 8(g), it is said that "The WVMR increases from the plume center to the edge, suggesting that the WVMR is an important role in the aging process." I don't see that increase in the WVMR profile in figure 8(g), rather that profile follows the profiles of $\beta$ and $\beta_F$.

Reply:  The sentence quoted was not well written. I wanted to express that the WVMR in the core of BBA plume, i.e. 5000-6000 m, was higher than at the edge 7000-9000 m  The sentence is rewritten as follows:

"The WVMR in the core of the plume is obviously higher than at the plume edge, suggesting that the WVMR is a potential indicator and/or factor in the aging process."

7. Line 231: "A sharp increase of PLDR$_{1064}$ to nearly 0.10 was detected at 8600 m, indicating the presence of ice crystals". Two remarks: a) Where should the reader look for that sharp increase? b) See my second remark in point 4.

Reply:  Reference to Figure 10(e) has been added.

8. Line 275: "The vertical variation of lidar ratios are also obvious in September (Figure 5)". First, it's unnecessary to talk about September, as all the presented measurements were carried out in September. Second, I don't see that fig. 5(a) makes obvious that variation. Although a variation in the lidar ratios can be seen between periods P1-P6 and periods P7-P9, I don't see that this variation is related to some vertical variation. For example, in period P2 the BBA layer is detected between 5500 and 8500 m, and in period P7, with lower lidar ratio, between 5400 and 8500 m, which is sensibly the same range as for P2. Please check if the figure referred to should be figure 5 or else be more explicit about what the reader should pay attention to.

Reply: (1) 'September removed'. (2) Yes, it shoud be Figure 5. The statistics shown in Figure 5 are taken from the plumes observed in different time interval. It means that, for one time interval, the data points in the box plot are simultaneously presented at consecutive vertical levels. And the vertical variations are reflected by the height of the boxes. In Figure 5(a), the vertical variations of lidar ratios are important, especially in P3 and P4.  For example, in P4, LR$_{532}$ varied from about 52 sr to 90 sr at 5500-7500 m.

9. Line 279: "From this aspect, the fluorescence capacity is a better parameter for aerosol typing especially in low aerosol concentrations". Please be more specific as to the meaning of "typing" in this context.

Reply: Typing means classification, to classify aerosol types into different categories, such as pollen, dust, smoke and so on.

**Formal remarks and typos**

1. In the bibliography, please insert the doi whenever possible.

   Reply:  doi numbers are now added to the bibliography.

2. As a general rule, I would prefer that the full wording of acronym is used before the acronym; for example, in line 68, I would prefer "showed enhanced particle linear depolarization ratios (PDLRs)" than the current "showed enhanced PLDRs (particle linear depolarization ratio)".

   Reply: corrected

3. Line 31: "They could alternate the planetary radiation budget of the planet". I think that the meaning is "They could alter the radiation budget of the planet".

   Reply: Yes, it should be alter.

4. Lines 80-81: "could influence the ability in water diffusion". This sentence sounds strange. Should it be "could influence the ability for water diffusion", or there is a word missing? Please check.

Reply: Yes, "the ability of water diffusion" is better.

5. Lines 96-97: ATOLL is defined (Atmospheric Observatory of LiLle), but is never used afterwards.

   Reply: Yes, after correction, ATOLL, instead of Lille, is refered when mentioning the lidar site.

6. Figs. 1(c) and 2 are not cited in the text. Also, the text where the call to figure 2 should be inserted (section 2.2) comes after the call to figure 3 in section 2.1, so the numbering of these figures should be swapped.

   Reply: corrected

7. Line 162: "and the height range of the layers are resented". "resented" should be "presented".

   Reply: corrected

8. Line 162: please insert the acronym LR after the lidar ratio is introduced, e.g. "Lidar ratio (LR), i.e. the ratio between extinction coefficient and backscatter coefficient".

   Reply: corrected

9. Lines 171-173: "Apart from the temporal variations, vertical variations in the BBA layers are also significant, such as lidar ratios in on 12 and 14 September, PLDRs on 11 and 18 September. Such variations are possibly indications of the variabilities in the burning materials, combustion conditions and aging process". Where are these lidar ratios and PLDRs represented? Where the reader should look at?

   Reply: The vertical variations in the BBA layers, indicated by the distance between the bottom and the top of each box plot in Figure 5, are also significant, such as lidar ratios in on 12 and 14 September, the PLDRs on 11 and 18 September.

10. Lines 188-189: "Figure 8 plots the parameters obtained from averaged observations between 22:00 UTC, 11 September and 03:00 UTC, 12 September". However, the caption of figure 8 says that the averaging end time is 02:00 UTC.

    Reply: Corrected. The caption was mistaken, it should be 03:00 UTC.

11. Line 197: "The increasing trend". Should it be "decreasing"? (see the sentence just before and fig. 8).

    Reply: Corrected. It should be "decreasing".

12. Line 205: Although evident for the expert, the acronyms WVMR and RH have not been defined.

    Reply: Definitions are added at the places where the two acronyms first appear.

13. Line 253: "episode of Canadian smoke over Europe". Please insert references, even if it implies repeating some of those already given in line 251.

    Reply: References added.

14. As general rule, increase the size of legends, labels and scales in the graphs. Be also explicit in the labels. For example, the label of the color bar of figure 4(b) should read "Volume depolarization ratio at 1064 nm" instead of "VDR_1064_AN". I don't know if specifying that measurements are obtained from analogue (AN) or photon counting (PC) channels is relevant in the context of this paper.

    Reply: The size of the labels have been increased. 'AN' and 'PC' are indeed unnecessary information and have been removed.

15. Figure 2: the labels (a) and (b) are missing in the figure panels.

    Reply: labels are added.

16. Figure 5. Please explain that P1, P2,...P9 in the horizontal scales refer to the periods identified in table 1.

    Reply: The explanation has been added.

17. Figure 5 caption, 3$^{rd}$ line: "The dot and bar in the box indicate the mean value and the middle value". "middle value" should probably be "median value".

    Reply: Corrected.

18. Although the English writing is very good in general, I would suggest a review by a native English speaker to fix some odd uses. Just as an example, in lines 12-13, the "varied" in the sentence "It reflects that the properties of aged BBA particles are highly varied" should probably be "variable"; in line 15 "than those" sounds better than "with those", etc.

    Reply: English writing has been improved.

---

## Author Comment (AC2)

**Reply to Review #2**

*by Qiaoyun HU*

The paper discusses interesting measurements of aged biomass burning smoke with a unique lidar. The paper is well written and appropriate for ACP. The measurements are performed with a recently introduced advanced lidar that combines multiwavelength lidar, Raman lidar, polarization lidar, and (new!) fluorescence lidar techniques.

Minor revisions are necessary.

p3, l75: When discussing INP, please keep in mind that these aged smoke particles are organic aerosol particles, the organic properties (of humic-like substances) count, and not the ones for soot or fly ash. Therefore, Knopf et al. 2018 …. is appropriate as reference.

Knopf, D. A., Alpert, P. A., and Wang, B.: The role of organic aerosol in atmospheric ice nucleation: a review, ACS Earth and Space Chemistry, 2, 168–202, https://doi.org/10.1021/acsearthspacechem.7b00120, 2018.

Reply: added.

p4, l96: One may cite Baars et al., 2021: Baars, H., et al. (2021). Californian wildfire smoke over Europe: A first example of the aerosol observing capabilities of Aeolus compared to ground-based lidar. Geophysical Research Letters, 48, e2020GL092194. https://doi.org/10.1029/2020GL092194

Reply: added.

p4, l105: Please use 'pyroCb' instead of 'pyCb'!

Reply: Corrected.

p4, l112: Figure 1(b) is mentioned, and then (l114) Figure 3 is mentioned. Figure 2 is left out.

Reply: The order of Figure 2 and 3 is swapped.

P5, l126: No one should introduce Figure 2!

Reply: Reference to Figure 2 added.

p5, l128: AE decreased…

Reply: Corrected. Yes, AE decreased.

p5, l133: To my understanding, Cimel (AERONET) is unable to correctly measure AODs>4.0. And now we have peak AODs of 5.8!

Reply: The new photometer is able to measure AOD up to 7.0. The AOD of 5.8 remains in the Level 2 data, which means it has been validated.

The Case study section is a bit boring, one should better emphasize the deviation of the optical properties on 17-18 Sep 2020 in Figure 5 from the rest, to make the entire story more exciting.

Reply: The presentation of case studies is now modified in the following ways:

1. In the Case 1, the introduction of BBA characteristics is kept and simplified.

2. In Case 2, the differences of BBA characteristics compared to Case 1 are emphasized.

So the repetition of numbers is avoided and the comparison of BBA properties is more clear.

Section 4: Discussion

I miss a clear structure of this section. The discussion could be shortened and should clearly highlight the added value now available in terms of the fluorescence information. Please state clearly: What is new! The discussion should be some kind of a review of the recent Veselovskii papers 2020 (general method), 2021 (on pollen) and the recent one on smoke/cirrus observations (also submitted in 2021) together with the present article on North American smoke.

I would leave out any speculation. For example, the discussion on age of smoke as a function of height. This is just speculation, and usually depends on many different factors such meteorological conditions, fire type, burning material, size of burning area and so on....).

Some suggestions that should be considered. The smoke particles are usually glassy in the upper troposphere and stratosphere (see the review article of Knopf et al.) The organic coating means that the INP properties are controlled by organic (humic-like) material. When discussing heterogeneous ice formation, do not restrict yourself to mixed phase clouds and temperatures higher than -35C. Heterogeneous ice nucleation also occurs at -50 to -70C (in cirrus). All this should be mentioned.

Furthermore, PLDR (or better, ... the shape properties) seem to depend on relative humidity (availability of water vapor) and further gases that can condense on smoke

particles to make them spherical. And the concentration of the gases are high in the lower troposphere and then obviously decrease with height from the middle to the dry upper troposphere and the extremely dry stratosphere.

page 9, line 267-274: I would leave out such a discussion.

Reply: The discussion has been condensed and restructured in the following ways:

1. Speculations and unnecessary discussions about modeling are removed.

2. Section 4.1 and 4.2 merged.

3. The discussion about smoke acting as INP shortened and condensed.

4. The discussion is organized in four paragraphs, each with one different topic: PLDR, lidar ratio, aerosol fluorescence and smoke acting as INP.

page 9, line 275-280, please state clearly how you calculate the lidar ratio, You cannot combine extinction and backscatter values obtained with DIFFERENT smoothing lengths.

Reply: After checking the code of lidar data processing, I confirm that the smoothing length for extinction and backscatter profiles is the same ! I forgot that I had considered the possible artifacts of using different smoothing length when developping the code longtime ago, so the statement in the manuscript was wrong and it has been removed. Thanks a lot for this remark !

page 9, line 287 to page 10, line 297: I would leave this discussion out as well. The paper deals with fluorescence. Please clearly state what is new...! Provide clear facts, what the added value is!

An extra section 4.2 on BBA as INP is not needed in this fluorescence-related paper. A paragraph on the impact of smoke serving as INP is sufficient, but please cover the full range of clouds from mixed phase clouds to cirrus (-25C to -70C), and then a reference to the recent Veselovskii paper on smoke-cirrus interaction is needed.

Veselovskii, I., Hu, Q., Ansmann, A., Goloub, P., Podvin, T., and Korenskiy, M.: Fluorescence lidar observations of wildfire smoke inside cirrus: A contribution to smoke-cirrus – interaction research, Atmos. Chem. Phys. Discuss. [preprint], https://doi.org/10.5194/acp-2021-1017, in review, 2021.

Reply: Section 4.1 and 4.2 have been merged.

The conclusion section should finally also be better organized and structured. I do not agree that the fluorescence information is the better information to identify

smoke. It is an additional one, more precise, another independent one, besides all the useful information on PLDR spectrum and lidar ratio spectrum.

Reply: The conclusion section has been re-organized. The sentence--"the fluorescence information is the better information to identify smoke" is ambiguous and has been removed from the conclusion and elsewhere. We want to emphasize that the fluorescence is very sensitive and is accessible even at low aerosol concentration and high altitude, while the calculation of lidar ratio requires smooth and at least moderate concentration of aerosols. But for sure the fluorescence is a supplementary information and should not replace extinction or backscattering measurements.

Figure 3: mixed-phase cloud at 10 km height? Impossible!

Reply: This mixed phase cloud was detected in a subtropical region at (37.91N, 85.41W) by CALIPSO. The temperature at this altitude was roughly about -40 degrees, but the uncertainty could be large and the occurrence of super-cooled liquid water is still possible. In Lille (50.6N, 3.1E), we observed supercooled liquid water clouds at 8 km height and mixed phase clouds at higher altitudes in September. The appearance of ice crystals can be confirmed with the depolarization ratio of 0.2--0.4. These ice crystals appeared in a smoke layer and were possibly initiated by smoke particles. But the nucleation pathway cannot be revolved with the information we currently have and CALIPSO did not observed any mixed phase cloud.  So I decided to changed "mixed phase cloud" into "ice crystals mixed with BBA" in order to avoid inaccurate expression.

[Figure]

*Fig 1. The depolarization ratio at 532 nm, CALIPSO measurements on 14-09-2020*

[Figure]

*Fig 2. The total attenuated backscatter at 532 nm, CALIPSO measurements on 14-09-2020 overlaid with temperature and potential temperature.*

Figure 4: All axis text must be enlarged, … is much too small at the moment.

Reply: Label size increased

Figure 5: (d) Y-axis EAE/BAE is confusing, better write …. EAE, BAE. In the caption, please explain explicitly which intense parameters are shown. What does P1-P9 mean? Please state that P1-P9 are listed in Table 1….

Reply:  Corrected.

Figure 7: Again, all axis text must be enlarged, much too small at the moment.

Reply:  Ticks are enlarged.

Figure 8: I would recommend to explain clearly what parameters are shown. Figures should be widely self-explaining.

Reply:  More explanations have been added.

Figure 9: Again, all axis text must be enlarged, much too small at the moment.

Reply:  Axis ticks are enlarged.

Figure 10: Here one could then state: Same as Figure 8, except…..

Reply:  Caption updated.

Figure 11: Again, what is shown…  should be stated.

Reply:  Caption updated.

Regarding all the figures, keep in mind that many readers may not be lidar specialists and need a lot of information.

Reply:  More information has been added in the caption of figures.